# Treatment Provided in Children and Adolescents with Functional Seizures—A Danish Nationwide Cohort

**DOI:** 10.3390/children10071218

**Published:** 2023-07-14

**Authors:** Nana Brandborg Sørensen, René Ernst Nielsen, Ann-Eva Christensen, Mikkel Kristian Kjeldsen, Charlotte Ulrikka Rask, Jakob Christensen, Anne Sofie Hansen

**Affiliations:** 1Unit for Psychiatric Research, Psychiatry, Aalborg University Hospital, 9000 Aalborg, Denmark; ren@rn.dk (R.E.N.); anneva.christensen@rn.dk (A.-E.C.); mikkel.kjeldsen@rn.dk (M.K.K.); ansoha@rn.dk (A.S.H.); 2Department of Clinical Medicine, Aalborg University, 9000 Aalborg, Denmark; 3Research Unit, Department of Child and Adolescent Psychiatry, Aarhus University Hospital, Psychiatry, 8200 Aarhus, Denmark; charrask@rm.dk; 4Department of Clinical Medicine, Aarhus University, 8200 Aarhus, Denmark; jakob@clin.au.dk; 5Department of Neurology, Aarhus University Hospital, 8200 Aarhus, Denmark

**Keywords:** functional seizures, psychogenic nonepileptic seizures, management, treatment, paediatric, Danish nationwide cohort

## Abstract

Background: Functional seizures (FS) are episodes of paroxysmal involuntary movements and altered consciousness without the typical changes in the electroencephalography as with epilepsy. A multidisciplinary approach is the golden standard in the treatment of FS. This study examined the cross-sectoral collaboration and treatment modalities provided to children and adolescents after a diagnosis of FS. Method: A Danish nationwide cohort, consisting of 334 children and adolescents, aged 5–17 years, with a validated diagnosis of FS during the period 2004–2014 was studied. Medical record data were collected from diagnosing hospital departments. Management and treatment modalities from the time of diagnosis up to three months after diagnosis were explored. Results: The most used treatment modalities were psychoeducation (*n* = 289, 86.5%) and follow-up in outpatient care (*n* = 192, 70.6%). A cross-sectoral collaboration was initiated for a third of cases (*n* = 98, 29.3%). The most commonly provided treatment combination consisted of psychoeducation, follow-up in outpatient care and psychotherapy; however, only a few patients received this specific combination (*n* = 14, 4.2%). Conclusions: The treatment applied was individualized and consisted of varying use of treatment modalities. Initiatives to curate clinical guidelines and implement a multidisciplinary treatment approach should be further explored to improve treatment for this young group of patients.

## 1. Introduction

The number of children and adolescents diagnosed with functional seizures (FS) has increased in recent years [1]. FS resemble epileptic seizures by their paroxysmal episodes of involuntary movements and altered consciousness. However, the semiology of FS differs from epileptic seizures and the electroencephalogram lacks the paroxysmal changes typically seen in patients with epilepsy [2,3,4]. The exact aetiology of FS is unknown, but prevailing explanatory models refer to interactions between various biological and psychosocial factors when describing the onset and triggers of FS [3,5]. Due to the similarity to epilepsy, patients diagnosed with FS often receive unnecessary interventions and potentially harmful treatment with antiseizure medication (ASMs), which can result in side effects without any efficacy against symptoms [2]. A multidisciplinary stepped care approach is regarded as the golden standard in the treatment of FS [4,5]. The aim of the approach is to help the patient and their family understand the nature of FS, how to control and contain FS and identify factors that could trigger or sustain FS [6]. In order to obtain the long-term goal of symptom remission without relapse, it is important to consult the patient’s professional and personal network to establish a cross-sectoral and cross-disciplinary understanding of the seizures and a collaborative treatment plan [7,8]. Studies have shown that one type of treatment modality might not be adequate for all patients, but a treatment guideline where it is possible to combine modalities designed for a single patient does show positive results [9]. To further aid in the curation of treatment guidelines, it is important to gain knowledge regarding current practice.

This study assesses the use of cross-sectoral collaboration and treatment modalities in the management of children and adolescents diagnosed with FS in a hospital setting.

## 2. Materials and Methods

### 2.1. Design

A retrospective Danish nationwide register- and population-based cohort study.

### 2.2. Population

This study utilized medical record data from a cohort of 386 children and adolescents, aged 5–17 years, diagnosed with FS during the period 1996 to 2014 in Denmark established by Hansen et al. The diagnosis of FS was validated using medical record data when creating the cohort [1].

### 2.3. The Study Population

Patients were excluded from the current study if the medical records did not contain notes from admission, release papers or the medical record did not supply data for the entire inpatient stay. Patients who received the diagnosis of FS prior to the year 2004 were excluded to examine the most recent clinical practice and possible time trends. Additionally, patients who received the diagnosis of FS at an emergency department or an internal medicine department were dismissed, since they solely provided diagnostic modalities and lacked a treatment plan to follow.

### 2.4. Outcomes

*Medical record data:* The medical record data contained data from three specialties: neurology, paediatrics and child and adolescent mental health services (CAMHS). Data were extracted from the medical records by the primary investigator (N.B.S.). A consensus meeting was conducted on a subsample of medical records prior to data extraction between N.B.S. and a senior medical doctor (A.S.H.) experienced within the field of child and adolescent psychiatry and FS. Based on the existing literature, 15 interventions were identified describing either treatment or management modalities, and these were assembled in a case report form used to extract data from the medical records [5,10,11,12,13]. N.B.S. consulted the senior medical doctor regarding data extraction in case of doubt to solve potential queries. The data were extracted from the time of the FS diagnosis up to three months following the FS diagnosis due to data availability, as most of the medical record data from the diagnosing departments did not extend this timeframe due to pragmatic reasons regarding the feasibility of the study. If the release date from inpatient care extended past the three month mark, the date of discharge was set as the end date of the available medical record data. Patients were included from the date they were informed about the FS diagnosis if diagnosed in outpatient care. Data were collected using the electronic data capture tool REDCAP hosted by the North Denmark Region [14].

Patient characteristics: The following data were extracted at the time of diagnosis: sex, age, time period of available medical record data and pharmaceutical treatment with ASMs or psychotropic drugs. Data for each patient regarding comorbid epilepsy, comorbid somatic illness and comorbid psychiatric disorders were gathered using data from the prior established cohort. 

Management and treatment modalities: The medical records were thoroughly read to assess the use of the 15 interventions describing either treatment or management modalities. Treatment modalities included (1) psychoeducation; (2) psychotherapy provided by a health care professional; (3) changes in pharmaceutical treatment with ASMs or psychotropic drugs; (4) guidance to school staff (i.e., knowledge and recommendations on how to respond to seizures in school); (5) network meetings (including a minimum of three professions such as social workers, school staff and medical staff) [13]; (6) inpatient school and pedagogical observation; and (7) physiotherapy (if used in the treatment of FS).

Management modalities included: (1) assessments performed at the department by other healthcare workers were carried out to explore whether they could offer other treatment modalities than the ones mentioned; (2) psychiatric liaison consultation; (3) contact with social services; and (4) pedagogical–psychological counselling (PPC). PPC in Denmark consists of pedagogues and psychologists who can aid in the identification of potential challenges these children might have, e.g., in a school setting, (5) follow-up in outpatient care (for patients who received the diagnosis of FS in inpatient care); (6) referral to another medical specialty; (7) discharge to the general practitioner (meaning the patient was discharged within the 3-month follow-up); (8) suggestions to the general practitioner (as described in the discharge notes).

Statistical analysis: The statistical analyses were conducted using STATA 17 [15]. Non-normally distributed continuous variables were presented by the median and interquartile range ([Q1, Q3]). Categorical variables were presented as frequencies and percentages. Comparisons of the distribution of management and treatment modalities across specialties were achieved using Fisher’s exact test. Stratified analyses on the groups “comorbid epilepsy”, “comorbid somatic illness” and “comorbid psychiatric disorder” were conducted.

## 3. Results

### 3.1. Patient Characteristics

The prior established cohort detected 386 patients with a diagnosis of FS [1]. A total of 334 patients were available for follow-up in the current study (Figure 1).

Patient characteristics are seen in Table 1. Most patients were female (82.6%). CAMHS presented the longest duration of medical record data (median 74 days, IQR [53.0–89.0]), whereas neurology and paediatrics reported shorter durations (median 48 days, IQR [8.0–73.0] and median 45 days, IQR [5.0–77.0], respectively). Those diagnosed in neurology departments were slightly older (median 16.8, IQR [16.3, 17.3]) than those diagnosed at paediatric departments (median 14.3 IQR [12.6, 15.3]) or at CAMHS (median 15.1 IQR [13.7, 16.2]). They also had a slightly higher rate of comorbid psychiatric disorders (23.2% vs. 17.6% at paediatrics and <18.5% at CAMHS) and of comorbid severe somatic illness (9.2% vs. 6.1% at paediatrics and 0% at CAMHS). Comorbid epilepsy was comparable among diagnosing departments. A total of 103 received pharmaceutical treatment at the time of diagnosis. Hereof, 81 received ASMs, 27 received psychotropic drugs and 5 received both. The rate of patients receiving ASMs was higher when the diagnosing department was neurology (84.0% of patients receiving pharmaceutical treatment at neurology), and the highest proportion of patients receiving psychotropic drugs was among patients diagnosed at CAMHS (25.5%). The FS diagnosis was given as inpatient most frequently in paediatric departments (89.7%).

### 3.2. Management and Treatment Modalities

The usage of management and treatment modalities is seen in Table 2. Psychoeducation was the modality most often applied (*n* = 289, 86.5%) and only seldom as a stand-alone treatment (*n* = 7, 2.1%). The second most frequent modality offered was follow-up in outpatient care, when diagnosed as an inpatient (*n* = 192 out of 272 diagnosed as an inpatient, 70.6%) with comparable application among specialties. Psychotherapy was the third most frequently offered modality, though still applied in less than half of the cases overall (*n* = 148, 44.3%). However, there was a significant difference in the application among specialties (*p* < 0.001) with CAMHS providing psychotherapy at a higher rate (*n* = 22, 81.5%) than paediatrics (*n* = 73, 44.2%) and neurology (*n* = 53, 37.3).

A combination of psychoeducation, psychotherapy, follow-up in inpatient care, inpatient school and pedagogical observation, and cross-sectoral collaboration was the most used combination for patients diagnosed in inpatient care (*n* = 15, 5.5%). Psychoeducation, follow-up in outpatient care if diagnosed as an inpatient and psychotherapy was the second most used combination for patients diagnosed during inpatient care (*n* = 14, 5.1%). Only a few patients received a combination of modalities that did not involve psychoeducation (*n* = 16, 4.8%). The application of psychoeducation was comparable among specialties. The combinations of treatment modalities received by five or more patients are visualized in Figure 2.

A cross-sectoral collaboration consisting of guidance to school staff (*n* = 66, 19.8%), contact with social services (*n* = 23, 6.9%) and network meetings (*n* = 22, 6.6%) was initiated for a third of the total study population (*n* = 98, 29.3%), the distribution of which was significantly different amongst diagnosing specialties (*p*-value < 0.001), with CAMHS showing a higher rate (*n* = 14, 51.85%) than paediatrics (*n* = 67, 40.6%) and neurology (*n* = 17, 12.0%). In 12 cases, a combination of 2 or more of these 3 modalities was applied. Change in the pharmaceutical treatment, i.e., either initiation or discontinuation of ASMs or psychotropic drugs, was applied to 51 patients (15.3%), with the highest fraction seen at CAMHS (*n* = 5, 18.5%) and neurology (*n*= 25, 17.6%). Treatment with psychotropic drugs was discontinued for less than 5 patients and initiated for 13 patients after the time of diagnosis of FS. Treatment with ASMs was initiated for a few patients after the diagnosis of FS was given. Referral to other medical specialties was carried out for a minority of patients at similar fractions for the specialties (total *n* = 79, 23.7%). CAMHS referred the majority of these to a tertiary specialized epilepsy clinic (*n* = 5, 71.4% of referrals from CAMHS), whereas neurology and paediatrics referred the majority to CAMHS (*n* = 20, 58.8% of referrals from neurology vs. *n* = 19, 50.0% of referrals from paediatrics).

In general, a combination of four modalities was applied to almost a quarter of the patients (*n* = 73, 21.9%); however, it was common to use a combination of two (*n* = 59, 17.7%), three (*n* = 71, 21.20%) or five (*n* = 66, 19.8%) modalities. The remaining patients received one modality (*n* = 33, 9.9%) or more than five modalities or none (*n* = 32, 9.6%), where no received treatment modality was rare. Most combinations were individualized and occurred with less than five patients receiving that combination (*n* = 36, 58.1% of patients diagnosed in outpatient care; *n* = 160, 58.8% of patients diagnosed in inpatient care).

### 3.3. Stratified Analyses

We found no statistically significant difference in modalities between those with and without comorbid epilepsy, respectively. However, treatment with AEDs was applied to 34 patients without the diagnosis of epilepsy at the time of diagnosis of FS. The treatment with AEDs was discontinued in 26 (72.2%) patients without comorbid epilepsy and 10 (27.8%) with comorbid epilepsy. A statistically significant difference (*p* < 0.001) was found in the modality, “assessment performed at hospital department”, between those without somatic comorbidity (*n* = 20 out of 311, 6.4%) and those with somatic comorbidity (*n* = 8 out of 23, 34.8%). For the patients with psychiatric comorbidity (*n* = 66, 19.8%) versus those without psychiatric comorbidity (*n* = 268, 80.2%), we found a significant difference in the application of psychoeducation (*n* = 49, 74.2% vs. *n*= 240, 89.6%, *p* = 0.002), change in pharmaceutical treatment (*n* = 16, 24.2% vs. *n*= 35, 13.1%, *p* = 0.034) and referral to medical specialty (*n* = 30, 45.5% vs. *n*= 49, 18.3%, *p* < 0.001).

## 4. Discussion

To our knowledge, this is the first study to examine the treatment provided to children and adolescents after a diagnosis of FS in a hospital setting on a nationwide basis. The current study found that most patients received psychoeducation, mostly in combination with other treatment modalities and less than half of the patients received psychotherapy. A cross-sectoral collaboration was rarely initiated. Most of the interventions were provided less to patients with comorbid psychiatric disorders than to patients without a comorbid psychiatric disorder, with significant differences found in the modalities psychoeducation, change in pharmaceutical treatment and referral to medical specialty.

### 4.1. The Multidisciplinary Approach and the Cross-Sectoral Collaboration

Contact with and activation of the patient’s network and social services is important to re-establish the daily life of the patients [9,11,12]. The literature describing the effects of contact with social services in the treatment of FS is sparse; however, the importance of social workers should not be neglected, since their gathering of the families’ social history can help uncover stressors in the patient’s daily lives [3,16]. Furthermore, social workers can facilitate follow-up appointments that could aid in furthering compliance with treatment as seen in the study by Terry et al. [17]. The study by Nielsen et al. examined the views on the management and treatment of FS in the paediatric population among 64 Danish paediatricians in 2013 [13]. Most paediatricians described that a multidisciplinary approach was applied “often”, general advice and psychoeducation were provided “always”, sessions with a psychologist were provided “often”, and referral to CAMHS, psychiatric consultations and physiotherapy were provided “sometimes” [13]. The current study shows that psychoeducation and follow-up in outpatient care indeed were provided to most patients, but the remaining services were used somewhat irregularly and the contact to CAMHS was sparse. This could indicate that there are inconsistencies in the treatment modalities that were reported by the paediatricians versus what treatment modalities were provided; however, the current study did only examine what modalities were provided in the first three months after diagnosis [13]. Additionally, the study by Nielsen et al. was a cross-sectional study examining the clinical practice in 2013, whereas the current study examines the trends during a 10-year period [13]. A publication by Caplan et al. suggests that, at a minimum, a team consisting of a mental health care professional, a paediatric neurologist, a general practitioner and school staff should be formed to ensure the optimal treatment for the patients [11].

Multiple observational studies have shown a decrease in FS frequency when a multidisciplinary approach is used [18,19,20]. The studies by Chandra et al., Kozlowska et al. and Chudleigh et al. are based at the same department providing the same 2-week inpatient mind-body program [18,21,22]. The creation of teams consisting of somatic healthcare specialists and mental healthcare specialists was a common denominator in the studies by Kozlowska et al., Chandra et al., Chudleigh et al., Flewelling et al. and McFarlane et al. [18,19,20,21,22]. Similarly, guidance to the patients’ schools was provided and stressors in the patients’ daily lives were identified. However, the treatment in the studies by Flewelling et al. and McFarlane et al. took place in an outpatient setting [19,20]. Kozlowska et al. found a greater acceptance of the diagnosis of FS and improved engagement in treatment at the 12-month follow-up, resulting in 75% of their 60 patients regaining normal function and full-time school attendance [18]. Flewelling et al. also found a decrease in seizure frequency, however, only for 46% of their 19 patients [20]. In total, 58% had improved school attendance and over 50% had a decrease in the usage of emergency departments. Both Flewelling et al. and McFarlane et al. did not find significant improvement regarding anxiety and depressive symptoms [19,20]. The greater positive outcome found in the studies by Kozlowska et al. could imply that focused inpatient care could potentially provide a quicker recovery for the patients; however, no scale was used to examine the level of anxiety or depressive symptoms in the patient group [18].

### 4.2. Psychoeducation and Psychotherapy

Psychoeducation, psychotherapy and reestablishment of the patients’ daily routines are important factors in the treatment of FS [11]. A number of studies have shown the positive effects of psychoeducation and psychotherapy in minimizing seizure frequency and improving daily function [9,18,23,24,25]. A study by Kozlowska et al. reported positive outcomes in seizure frequency when using mind-body psychotherapy; similar positive effects have also been reported by Fobian et al. using retraining and control psychotherapy [18,24]. Despite these results, psychotherapy was provided to less than 50% of the study population in the current study. It should be noted that not all departments had a psychologist or psychiatrist available, yet only a minority of patients were suggested to obtain a referral to a psychologist via their general practitioner. This could potentially lead to a delay in treatment and consequently a slower recovery. It would be beneficial to curate early referrals to mental health specialists such as psychologists or psychiatrists, either in the primary or secondary sector, who has the relevant skill set and knowledge to treat FS. This would, in turn, contribute to the multidisciplinary approach.

### 4.3. Pharmaceuticals

Most patients without comorbid epilepsy discontinued ASMs; however, ASMs were either initiated or continued for 11 (3.8%) patients without comorbid epilepsy. Multiple studies have shown that treatment with ASMs is ineffective and can potentially lead to adverse learning and behavioural effects in patients without epilepsy [4,26,27,28]. The results from the current study could reflect the results found in the study by Plioplys et al. [29]. This study suggests that the diagnostic uncertainty by the medical staff contributes to the incorrect treatment of ASMs in patients diagnosed with FS [29]. Psychotropic drugs should be used carefully but can be used to stabilize other symptoms, improving the foundation for psychotherapy as described by Chandra et al. [21].

### 4.4. Strengths and Limitations

The strengths of this study are the large sample size, long study period, good representativity and population-based design, making it possible to include medical record data from both regular hospital departments, but also tertiary specialized epilepsy clinics. The data were collected using a data retraction template, with pre-defined categories, to ensure reproducibility. Still, some limitations were present.

The current study is a retrospective study and, thus, relies on the reporting by health care professionals. Insufficient reporting could lead to biases or incomplete data. In addition, the variation in the level of detail in reporting made it difficult to assess the quality of the used modalities. Since the data was extracted by a single investigator, there is a possibility of human error and subjectivity in the data collection and interpretation. The medical records only contained data from the diagnosing department, meaning any service provided at another hospital department, e.g., after a referral or at the general practitioner’s would not be available for review in the current study. Modalities used in the treatment of FS after the first three months post diagnosis of FS were not registered in the current study. The data extracted from the medical records in the current study are from the past decade hence the results might not demonstrate current practices. Nevertheless, it is important to have knowledge of previous practices to curate future treatment guidelines. Furthermore, the layout of the healthcare system is different from country to country, which is why the results found in this study might differ from the reality in other countries. This limits the generalizability of the study and possibly affects the applicability of the findings in the current study to other populations and healthcare systems.

Limitations that were present in the creation of the cohort could lower the validity of the diagnosis of FS and possibly cause a more highly selected group of paediatric patients diagnosed with FS, as described in the study by Hansen et al. [1].

## 5. Conclusions

To our knowledge, this is the first study to examine the nationwide clinical practice of treatments for FS across medical specialties for children and adolescents. Psychoeducation and follow-up in outpatient care were provided as a treatment to most patients, whereas a cross-sectoral collaboration and a multidisciplinary approach were rarely provided, though the literature has described these as recommended interventions. These findings suggest that initiatives to develop clinical guidelines that support the implementation of a cross-sectoral and multidisciplinary approach are warranted to optimize the treatment provided to this young group of patients.

## Figures and Tables

**Figure 1 children-10-01218-f001:**
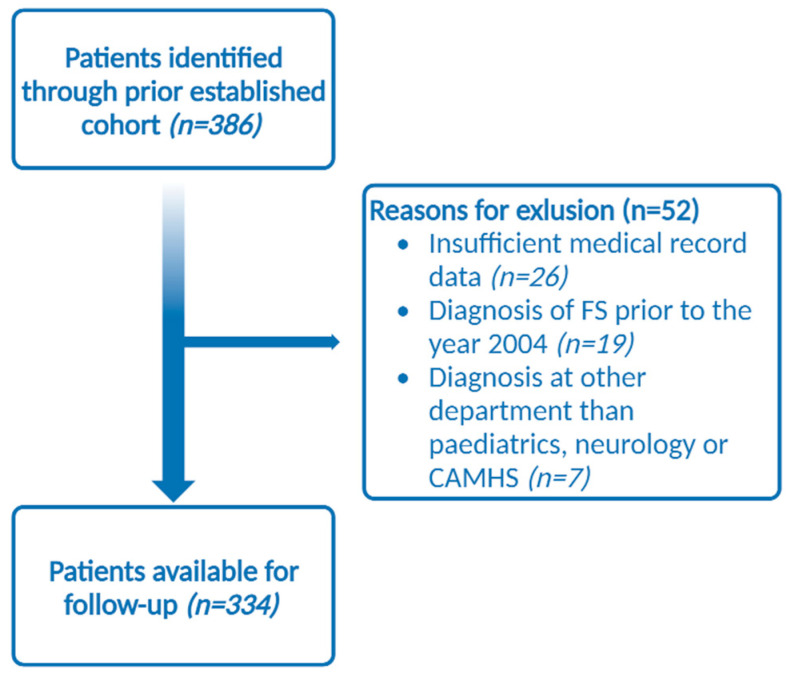
Flowchart of the detection of patients available for follow-up.

**Figure 2 children-10-01218-f002:**
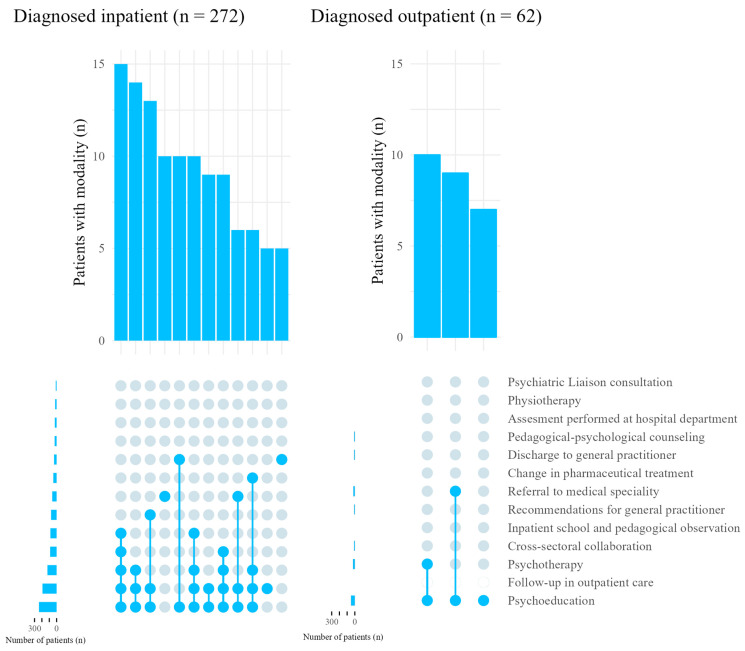
Combinations of treatment and management modalities used for five or more patients diagnosed either in inpatient care or outpatient care. Patients diagnosed in outpatient care could not receive the modality follow-up in outpatient care after the diagnosis of FS, since they received this modality prior to the diagnosis of FS. Combinations that appeared less than five times are not visualized in the plot. Modalities used in the combination are marked by a dot. The number of patients who received the specific modality is shown on the left of the dots representing the different modalities. The number of patients who received the specific combinations is shown above the dots representing the different modalities.

**Table 1 children-10-01218-t001:** Patient characteristics for children and adolescents diagnosed with FS in regard to diagnosing specialty ^a^.

	Total *N* = 334	Neurology *n* = 142	Paediatrics *n* = 165	CAMHS *n* = 27
Female sex	276 (82.6)	123 (86.6)	132 (80.0)	21 (77.8)
Age at diagnosis (Median, [IQR]) ^b^	15.7 [14.1–16.8]	16.8 [16.3–17.3]	14.3 [12.6–15.3]	15.1 [13.7–16.2]
Duration of medical record data in days (median, [IQR]) ^b^	49.0 [8.0–78.0]	48.0 [8.0–73.0]	45.0 [5.0–77.0]	74.0 [53.0–89.0]
Comorbid epilepsy ^c^	54 (19.2)	<30	27 (16.4)	<5
Comorbid severe somatic illness ^c^	23 (6.9)	13 (9.2)	10 (6.1)	0 (0.0)
Comorbid psychiatric disorders ^c^	<67 (<20.1)	33 (23.2)	29 (17.6)	<5
Pharmaceutical treatment at time of diagnosis	103 (30.8) ^d^	50 (35.2)	47 (28.5)	6 (22.2)
ASMs ^e^	81 (78.6)	42 (84.0)	>34 (>72.3)	<5
Psychotropic drugs ^e^	27 (26.2)	>10 (>20.0)	12 (25.5)	<5
Diagnosis given as inpatient	271 (81.4)	112 (78.9)	148 (89.7)	11 (40.7)

^a^ Data are presented as frequency (percentage) unless otherwise indicated. Subcategories are presented as frequency (percentage of main category). Due to Danish data protection rules, numbers above zero, but below five, are set to <5. Furthermore, the number for the total study population for the categories is adjusted accordingly. The total study population will, in these instances, be set to < calculated study population. ^b^ IQR depicts the 25th and 75th percentile. ^c^ Data used to calculate the number of patients diagnosed with comorbid epilepsy, severe somatic illness or psychiatric disorders was retrieved via medical record registries from the prior established cohort by Hansen et al. [1] ^d^ Describes the number of patients who received pharmaceutical treatment with either ASMs, psychotropic drugs or both. ^e^ Number includes the 5 patients who received treatment with both ASMs and psychotropic drugs at time of diagnosis.

**Table 2 children-10-01218-t002:** Management and treatment modalities for children and adolescents diagnosed with FS in regard to diagnosing specialty ^a^.

	Total *N* = 334	Neurology *n* = 142	Paediatrics *n* = 165	CAMHS *n* = 27	*p*-Value ^b^
Psychoeducation	289 (86.5)	121 (85.2)	146 (88.5)	22 (81.5)	0.46
Psychotherapy	148 (44.3)	53 (37.3)	73 (44.2)	22 (81.5)	<0.001
Change in pharmaceutical treatment	51 (15.3)	25 (17.6)	21 (12.7)	5 (18.5)	0.44
Cross-sectoral collaboration ^c^	98 (29.3)	17 (12.0)	67 (40.6)	14 (51.9)	<0.001
Inpatient school and pedagogical observation	85 (25.4)	23 (16.2)	62 (37.6)	0 (0.0)	<0.001
Physiotherapy	23 (6.9)	<5	17 (10.3)	<5	<0.001
Assessment performed at hospital department	<32 (<9.6)	16 (11.3)	11 (6.7)	<5	0.32
Psychiatric liaison consultation	<15 (<4.5)	10 (7.0)	<5	0 (0.0)	0.052
Pedagogical–psychological counselling	<38 (<11.4)	<5	23 (13.9)	10 (37.0)	<0.001
If inpatient diagnosis (*n* = 272): follow-up in outpatient care	192 (70.6)	76 (67.9)	107 (71.8)	9 (81.8)	0.59
Referral to medical specialty	79 (23.7)	34 (23.9)	38 (23.0)	7 (25.9)	0.92
Discharge to general practitioner ^d^	<47 (<14.1)	18 (12.7)	24 (14.5)	<5	0.71
Suggestions for general practitioner	<86 (<25.7)	39 (27.5)	42 (25.5)	<5	0.13

^a^ Data are presented as frequency (percentage) unless otherwise indicated. Due to Danish data protection rules, numbers above zero, but below five are set to <5 and the number of the total study population for these categories will be adjusted if one specialty reports <5 in a category, where <5 is counted as 5. Here, the total study population will be set to < calculated study population. Each patient could receive more than one management or treatment modality. ^b^ *p*-values are based on Fisher’s exact test. ^c^ The category consists of the following modalities: guidance to school staff, network meetings and contact with social services. Network meetings consisted of at least three different professions, such as social workers, school staff, general practitioners or health care professionals from the department. Inspired by Nielsen et al. [13]. ^d^ Describes the number of patients discharged in the 3-month period after the diagnosis of FS.

## Data Availability

The data presented in this study are available on request from the corresponding author. The data are not publicly available due to GDPR regulations.

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
