# Peer review of "Treatment Provided in Children and Adolescents with Functional Seizures—A Danish Nationwide Cohort"

_children, 2023, doi:10.3390/children10071218_

Round 1

Reviewer 1 Report

The study investigated the treatment provided to children and adolescents diagnosed with functional seizures (FS) in a nationwide hospital setting. The manuscript is well-crafted, and the research question is clearly articulated.

Advantages:

·       Nationwide Scope: The study encompassed a nationwide cohort of children and adolescents diagnosed with functional seizures, offering a comprehensive overview of the treatment administered in a hospital setting.

·       Multidisciplinary Approach: The study underscored the significance of adopting a multidisciplinary approach in the treatment of functional seizures, involving neurologists, mental health professionals, and other clinicians. This highlights the holistic care provided to patients.

·       Real-World Data: By utilizing medical records, the study captured real-world treatment practices and patterns for functional seizures, providing insights into the actual clinical management of patients in a hospital setting.

·       Identification of Treatment Modalities: The study identified the most frequently employed treatment modalities, including psychoeducation, psychotherapy, and follow-up care. This information can serve as a guide for healthcare providers in designing treatment plans for patients with functional seizures.

Weaknesses:

·       Retrospective Study Design: The study employed a retrospective design, relying on data extracted from medical records. This design limits the researchers' control over data collection and may introduce biases or incomplete data.

·       Limited Generalizability: The study was conducted in Denmark, which may have specific healthcare practices and resources that differ from other countries. Consequently, the findings may have limited applicability to other populations and healthcare systems.

·       Data Extraction by a Single Investigator: The primary investigator extracted data from medical records based on existing literature, which introduces the possibility of human error and subjectivity in data collection and interpretation.

Overall, the study highlights the importance of implementing a multidisciplinary approach in the treatment of functional seizures, with psychoeducation and follow-up in outpatient care emerging as the most commonly employed modalities.

Suggestions:

1.      Please include the weaknesses mentioned in my previous comments in the "limitations" section of the article.

2.      In the discussion section, consider providing recommendations on how to facilitate the early identification of functional seizures by professionals in neurology and pediatrics, and advocate for early referrals to mental health specialists for psychoeducation, psychotherapy, and interdisciplinary, team-based treatment.

Author Response

Thank you for the very kind comments and constructive feedback concerning our manuscript!

Reviewer #1 comment #1

The study investigated the treatment provided to children and adolescents diagnosed with functional seizures (FS) in a nationwide hospital setting. The manuscript is well-crafted, and the research question is clearly articulated.

Advantages:

  • Nationwide Scope: The study encompassed a nationwide cohort of children and adolescents diagnosed with functional seizures, offering a comprehensive overview of the treatment administered in a hospital setting.
  • Multidisciplinary Approach: The study underscored the significance of adopting a multidisciplinary approach in the treatment of functional seizures, involving neurologists, mental health professionals, and other clinicians. This highlights the holistic care provided to patients.
  • Real-World Data: By utilizing medical records, the study captured real-world treatment practices and patterns for functional seizures, providing insights into the actual clinical management of patients in a hospital setting.
  • Identification of Treatment Modalities: The study identified the most frequently employed treatment modalities, including psychoeducation, psychotherapy, and follow-up care. This information can serve as a guide for healthcare providers in designing treatment plans for patients with functional seizures.

Weaknesses:

  • Retrospective Study Design: The study employed a retrospective design, relying on data extracted from medical records. This design limits the researchers' control over data collection and may introduce biases or incomplete data.
  • Limited Generalizability: The study was conducted in Denmark, which may have specific healthcare practices and resources that differ from other countries.
  • Consequently, the findings may have limited applicability to other populations and healthcare systems.
  • Data Extraction by a Single Investigator: The primary investigator extracted data from medical records based on existing literature, which introduces the possibility of human error and subjectivity in data collection and interpretation.

Overall, the study highlights the importance of implementing a multidisciplinary approach in the treatment of functional seizures, with psychoeducation and follow-up in outpatient care emerging as the most commonly employed modalities.

Suggestions:

  1. Please include the weaknesses mentioned in my previous comments in the "limitations" section of the article.
  2. In the discussion section, consider providing recommendations on how to facilitate the early identification of functional seizures by professionals in neurology and pediatrics, and advocate for early referrals to mental healthspecialists for psychoeducation, psychotherapy, and interdisciplinary, team-based treatment.

Response:

Regarding suggestion #1: Thank you for the suggestions. These have been added.

Weakness #1: We fully agree that the retrospective study design relying on medical record data could present potential biases or incomplete data. We originally described this concern in the limitations section, but we have now expanded the wording according to your suggestion.

Revision:

“The current study is a retrospective study, and thus relies on the reporting by health care professionals. Insufficient reporting could lead to biases or incomplete data. In addition, the variation in the level of detail in reporting made it difficult to assess the quality of the used modalities.”

Weakness #2 and #3:  We are in agreement. We tried to word this in the limitations section, but have expanded the wording to include your suggestions.

Revision:

“Furthermore, the layout of the healthcare system is different from country to country, why the results found in this study might differ from the reality in other countries. This limits the generalizability of the study and possibly affects the applicability of the findings in the current study to other populations and healthcare systems.”

Weakness #4: Thank you for the valid suggestion/concern. The limitation have been added in the limitations section.

Revision:

“Since the data was extracted by a single investigator, there is a possibility of human error and subjectivity in the data collection and interpretation.”

Regarding suggestion #2: The issue on how to identify functional seizures is indeed very relevant in order to facilitate a relevant treatment plan for functional seizures. However, the scope of the current study is not to provide information about the early identification of functional seizures, why the information would not be applicable for the current study. We agree that it would be beneficial for future guidelines to contain both information about how to identify functional seizures and suggestions for how a multidisciplinary treatment plan could be curated.
    We agree that an interdisciplinary and team-based approach is preferable, including early referrals to mental health specialists for psychoeducation and -therapy. We have expanded the wording in the manuscript.

Revision:

“It would be beneficial to curate early referrals to mental health specialists such as psychologists or psychiatrists, either in the primary or secondary health care sector, who has the relevant skill set and knowledge to treat FS. This would, in turn, contribute to the multidisciplinary approach.”

General revisions:

“Statistical” was changed to “statistic” in two instances in the results section under 3.4 Stratified analysis.

“Assessment” was changed to “assessment in one instance in the results section under 3.4 Stratified analysis.

“Comobidity” was changed to “comorbidity” in one instance in the results section under 3.4 Stratified analysis.

“Speciality” was changed to “specialty” in one instance in the results section under 3.4 Stratified analysis.

Reviewer 2 Report

1.  What is the main question addressed by the research?

The authors have done a wonderful job of writing this manuscript on functional seizures.

This study conducted by Sorensen et al has determined the cross sectoral collaboration and the treatment modalities provided to children and adolescents after the diagnosis of functional seizures. A multidisciplinary team approach is expected to help patients and their families to understand the nature of functional seizures, ways to manage it, and also to identify the triggers.

This paper is aimed at gathering additional knowledge related to current  

 clinical practice of functional seizures so that treatment guidelines can be

 further curated

2. Do you consider the topic original or relevant in the field? Does it address a specific gap in the field?

Yes. The paper addresses the specific gap in the field about how patients with    

functional seizures were treated

3. What does it add to the subject area compared with other published

material?

This manuscript adds to the depth of how the patients with functional seizures were managed with multifactorial approach.

4. What specific improvements should the authors consider regarding the

methodology? What further controls should be considered?

Fig 2 needs more illustrations and legend explanations. Its is not conveying the message directly to the readers. The axis for the graphs pertaining to inpatient and out patient needs to be explained.

5. Are the conclusions consistent with the evidence and arguments presented

and do they address the main question posed?

Yes. But some attempts can be made about laying the clinical guidelines to treat functional seizures based on this study.

The Authors should mention why a long term follow up was either not attempted or was not feasible with this specific cohort. Considering the actual length of the study.

The result section can be aided with more graphs. Especially the two tables can be represented with several pie charts to be better retained in the memory of readers.

Author Response

Thank you for your kind comments and constructive feedback concerning our manuscript!

Reviewer #2

  1. What is the main question addressed by the research?
    The authors have done a wonderful job of writing this manuscript on functional seizures.
    This study conducted by Sorensen et al has determined the cross sectoral collaboration and the treatment modalities provided to children and adolescents after the diagnosis of functional seizures. A multidisciplinary team approach is expected to help patients and their families to understand the nature of functional seizures, ways to manage it and also to identify the triggers.
    This paper is aimed at gathering additional knowledge related to current clinical practice of functional seizures so that treatment guidelines can be further curated
  2. Do you consider the topic original or relevant in the field? Does it address a specific gap in the field?
    The paper addresses the specific gap in the field about how patients with functional seizures were treated
  3. What does it add to the subject area compared with other published material?
    This manuscript adds to the depth of how the patients with functional seizures were managed with multifactorial approach.
  4. What specific improvements should the authors consider regarding the methodology? What further controls should be considered?
    Fig 2 needs more illustrations and legend explanations. It is not conveying the message directly to the readers. The axis for the graphs pertaining to inpatient and outpatient needs to be explained.
  5. Are the conclusions consistent with the evidence and arguments presented and do they address the main question posed?
    But some attempts can be made about laying the clinical guidelines to treat functional seizures based on this study.
    The Authors should mention why a long term follow up was either not attempted or was not feasible with this specific cohort. Considering the actual length of the study.

Response:

Regarding comment #4: The suggestion has been taken into account and the figure and figure explanation have been revised.

Revision:

Figure 2. Combinations of treatment and management modalities used for five or more patients diagnosed either inpatient care or in outpatient care. Patients diagnosed in outpatient care could not receive the modality follow-up in outpatient care after the diagnosis of FS, since they received this modality prior to the diagnosis of FS. Combinations that appeared less than five times are not visualized in the plot. Modalities used in the combination are marked by a dot. The number of patients who received the specific modality is shown on the left to the dots representing the different modalities. The number of patients who received the specific combinations is shown above the dots representing the different modalities.“

Regarding comment #5:

Thank you for the constructive feedback and valid suggestions. We agree that the current study can aid in the curation of future clinical guidelines in the treatment of FS, however, we find it  important to examine the effect of the current practice on both seizure frequency and patient treatment satisfaction before such guidelines can be curated. Formulating clinical guidelines was not in the scope of the current study since the aim of the study was to outline the existing practice regarding use of cross-sectoral collaboration and treatment modalities.
   There was a time-limit of 5 months for the data extraction, analysis, interpretation and writing of the project, it was not possible to extend the inclusion period. Additionally, most of the medical record data ended shortly after the 3-month mark, hence the lack of availability of data. We have expanded the wording under the “Materials and methods” section under “Outcomes”.

Revision:

“The data were extracted from the time of the FS diagnosis up to three months following the FS diagnosis due to data availability, as most of the medical record data from the diagnosing departments did not extend this timeframe, and due to pragmatic reasons regarding the feasibility of the study.”

General revisions:

“Statistical” was changed to “statistic” in two instances in the results section under 3.4 Stratified analysis.

“Assessment” was changed to “assessment in one instance in the results section under 3.4 Stratified analysis.

“Comobidity” was changed to “comorbidity” in one instance in the results section under 3.4 Stratified analysis.

“Speciality” was changed to “specialty” in one instance in the results section under 3.4 Stratified analysis.

Thank you for your kind comments and constructive feedback concerning our manuscript!

Reviewer #2

  1. What is the main question addressed by the research?
    The authors have done a wonderful job of writing this manuscript on functional seizures.
    This study conducted by Sorensen et al has determined the cross sectoral collaboration and the treatment modalities provided to children and adolescents after the diagnosis of functional seizures. A multidisciplinary team approach is expected to help patients and their families to understand the nature of functional seizures, ways to manage it and also to identify the triggers.
    This paper is aimed at gathering additional knowledge related to current clinical practice of functional seizures so that treatment guidelines can be further curated
  2. Do you consider the topic original or relevant in the field? Does it address a specific gap in the field?
    The paper addresses the specific gap in the field about how patients with functional seizures were treated
  3. What does it add to the subject area compared with other published material?
    This manuscript adds to the depth of how the patients with functional seizures were managed with multifactorial approach.
  4. What specific improvements should the authors consider regarding the methodology? What further controls should be considered?
    Fig 2 needs more illustrations and legend explanations. It is not conveying the message directly to the readers. The axis for the graphs pertaining to inpatient and outpatient needs to be explained.
  5. Are the conclusions consistent with the evidence and arguments presented and do they address the main question posed?
    But some attempts can be made about laying the clinical guidelines to treat functional seizures based on this study.
    The Authors should mention why a long term follow up was either not attempted or was not feasible with this specific cohort. Considering the actual length of the study.

Response:

Regarding comment #4: The suggestion has been taken into account and the figure and figure explanation have been revised.

Revision:

Figure 2. Combinations of treatment and management modalities used for five or more patients diagnosed either inpatient care or in outpatient care. Patients diagnosed in outpatient care could not receive the modality follow-up in outpatient care after the diagnosis of FS, since they received this modality prior to the diagnosis of FS. Combinations that appeared less than five times are not visualized in the plot. Modalities used in the combination are marked by a dot. The number of patients who received the specific modality is shown on the left to the dots representing the different modalities. The number of patients who received the specific combinations is shown above the dots representing the different modalities.“

Regarding comment #5:

Thank you for the constructive feedback and valid suggestions. We agree that the current study can aid in the curation of future clinical guidelines in the treatment of FS, however, we find it  important to examine the effect of the current practice on both seizure frequency and patient treatment satisfaction before such guidelines can be curated. Formulating clinical guidelines was not in the scope of the current study since the aim of the study was to outline the existing practice regarding use of cross-sectoral collaboration and treatment modalities.
   There was a time-limit of 5 months for the data extraction, analysis, interpretation and writing of the project, it was not possible to extend the inclusion period. Additionally, most of the medical record data ended shortly after the 3-month mark, hence the lack of availability of data. We have expanded the wording under the “Materials and methods” section under “Outcomes”.

Revision:

“The data were extracted from the time of the FS diagnosis up to three months following the FS diagnosis due to data availability, as most of the medical record data from the diagnosing departments did not extend this timeframe, and due to pragmatic reasons regarding the feasibility of the study.”

General revisions:

“Statistical” was changed to “statistic” in two instances in the results section under 3.4 Stratified analysis.

“Assessment” was changed to “assessment in one instance in the results section under 3.4 Stratified analysis.

“Comobidity” was changed to “comorbidity” in one instance in the results section under 3.4 Stratified analysis.

“Speciality” was changed to “specialty” in one instance in the results section under 3.4 Stratified analysis.

Reviewer 3 Report

This manuscript presents the findings of a study that focused on the cross-sectoral collaboration and treatment modalities provided to children and adolescents diagnosed with functional seizures (FS). The study involved a Danish nationwide cohort of 334 individuals aged 5-17 years with a validated diagnosis of FS between 2004 and 2014. The researchers collected medical record data from diagnosing hospital departments and examined the management and treatment modalities from the time of diagnosis up to three months after diagnosis.

The results of the study indicate that the most used treatment modalities for FS were psychoeducation (86.5%) and follow-up in outpatient care (70.6%). A cross-sectoral collaboration, involving collaboration between different sectors or disciplines, was initiated for approximately one-third of the cases (29.3%). However, only a small number of patients (4.2%) received a specific treatment combination consisting of psychoeducation, follow-up in outpatient care, and psychotherapy.

Based on these findings, the authors conclude that the treatment provided for FS was individualized and involved varying use of treatment modalities. They suggest that further exploration of initiatives to develop clinical guidelines and implement a multidisciplinary treatment approach is necessary to improve the treatment of this group of young patients.

Overall, the study highlights the importance of a multidisciplinary approach in treating functional seizures and emphasizes the need for standardized treatment guidelines to enhance patient care. However, I have some questions:

What were the main treatment modalities provided to children and adolescents diagnosed with FS in the nationwide study?

Were there significant differences in the treatment provided to patients with comorbid psychiatric disorders compared to those without comorbidities?

How did the study emphasize the importance of social workers in the treatment of FS? What role did they play?

What recommendations did the study make regarding the formation of treatment teams and the involvement of various healthcare professionals?

What were the reported outcomes of studies that used a multidisciplinary approach, psychoeducation, and psychotherapy in the treatment of FS?

Why were antiseizure medications (ASMs) prescribed to some patients without comorbid epilepsy? What were the potential concerns associated with this practice?

What were the limitations of the study in terms of data collection and reporting? How might these limitations affect the validity of the findings?

What are the implications of the study's findings and recommendations for future treatment guidelines and clinical practice in the field of FS?

Minor points:

Correct the word “behavioural” in the line 304.

Correct the word “behavioural” in the line 304.

Author Response

Thank you for the kind comments and follow-up questions.

Reviewer #3 comment #1

What were the main treatment modalities provided to children and adolescents diagnosed with FS in the nationwide study?

We find that this question has been substantially answered in the results section under 3.2 Management and treatment modalities. The most used modalities have been listed after each other to further aid the reader in identifying these.

Revision:

“Psychoeducation was the modality most often applied (n=289, 86.5%) and only seldom as a stand-alone treatment (n=7, 2.1%). The second most frequent modality offered was follow-up in outpatient care, when diagnosed inpatient (n=192 out of 272 diagnosed inpatient, 70.6%) with comparable application among specialties. Psychotherapy was the third most frequent modality, though still applied in less than half of the cases overall (n=148, 44.3%). However, there was a significant difference in the application among specialties (p<0.001) with CAMHS providing it in a higher rate (n=22, 81.5%) than paediatrics (n=73, 44.2%) and neurology (n=53, 37.3).”

Reviewer #3 comment #2

Were there significant differences in the treatment provided to patients with comorbid psychiatric disorders compared to those without comorbidities?

We find that this question has been substantially answered in the results section under 3.4 Stratified analysis, we refer to the sentence: “For the patients with psychiatric comorbidity (n=66, 19.8%) versus those without psychiatric comorbidity (n=268, 80.2%), we found a significant difference in the application of psychoeducation (n=49, 74.2% vs. n= 240, 89.6%, p=0.002), change in pharmaceutical treatment (n=16, 24.2% vs. n= 35, 13.1%, p=0.034) and referral to medical specialty (n=30, 45.5% vs. n= 49, 18.3%, p<0.001).”

Reviewer #3 comment #3

How did the study emphasize the importance of social workers in the treatment of FS? What role did they play?

Thank you for your question. The scope of the current study was to outline the use of cross-sectoral collaboration and treatment modalities in the three months after diagnosis of functional seizures in children and adolescents. Thus, the scope of the study was not to examine the role nor importance of social workers in the treatment plan, however, this has been sparsely described in the literature, as described in the current study. We refer to the following sentences from the study:

“Contact to and activation of the patient´s network and social services is important to reestablish the daily life of the patients [11], [13], [14]. Literature describing the effects of contact to the social services in the treatment of FS is sparse, however, the importance of social workers should not be neglected, since their gathering of the families’ social history can help uncover stressors in the patients’ daily live [3], [18]. Furthermore, social workers can facilitate follow-up appointments that could aid further in the compliance to treatment as seen in the study by Terry et al. [19].

The following references were used:

[3] Asadi-Pooya A. A., Bahrami, Z. Social aspects of life in patients with functional seizures: Closing the gap in the biopsychosocial formulation. Epilepsy Behav. 117, 119-121, (2018) doi: 10.1016/j.yebeh.2021.107903. 

[11] Velani, H., Gledhill, J. The effectiveness of psychological interventions for children and adolescents with non-epileptic seizures. Seizure. 93. 20–31, (2021) doi: 10.1016/j.seizure.2021.09.006. 

[13] Caplan, R., Doss, J., Plioplys, S., Jones, J. E. Short-Term Treatment in Pediatric Psychogenic Non-Epileptic Seizures, 27-48 (Springer International Publishing, 2017)  

[14] Caplan, R., Doss, J., Plioplys, S., Jones, J. E. Long-Term Treatment in Pediatric Psychogenic Non-Epileptic Seizures (Caplan, R., Doss, J., Plioplys, S., Jones, J. E. eds) 49-62. (Springer International Publishing, Switzerland 2017). 

[18] Thompson, N. C., Gibson, P. A. Models of care: the roles of nurses and social workers in the diagnosis and management of patients with nonepileptic seizures in Gates and Rowan’s Nonepileptic Seizures, 3rd edn (Schacter, S. C., LaFrance Jr, W. C. Eds.) 253258 (Cambridge University Press, 2010). 

[19] Terry, D., Enciso, L., Trott, K., Burch, M., Albert, D. V. F. Outcomes in children and adolescents with psychogenic nonepileptic events using a multidisciplinary clinic approach. Jour. Of Child Neuro. 35, 918-923 (2020) doi: 10.1177/0883073820939400

Reviewer #3 comment #4

What recommendations did the study make regarding the formation of treatment teams and the involvement of various healthcare professionals?

The scope of the current study was not to curate clinical guidelines/recommendations for the treatment of functional seizures, however the recommendations from the existing literature were outlined. We refer to the introduction section of the study.

“In order to obtain the long-term goal of symptom remission without relapse, it is important to consult the patients professional and personal network to establish a cross-sectoral and cross-disciplinary understanding of the seizures and a collaborative treatment plan [9], [10].”

The following references were used:

[9] Heyman, I., Reilly, C. Seize the opportunity – Recognition and management of functional seizures in children. Eur. J. Paediatr. Neurol, 22, 734–735 (2018), doi: 10.1016/j.ejpn.2018.07.009.

[10] Rask C.U., Bonvanie I.J., Garralda E.M. Risk and Protective Factors and Course of Functional Somatic Symptoms in Young People. In: Understanding Uniqueness and Diversity in Child and Adolescent Mental Health. (2018).

We also refer to the results section under 3.2 Management and treatment modalities. We found that:

“A cross-sectoral collaboration consisting of guidance to school staff (n=66, 19.8%), contact to social services (n=23, 6.9%) and network meetings (n=22, 6.6%), was initiated for a third the total study population (n=98, 29.3%), the distribution of which was significantly different amongst diagnosing specialties (p-value < 0.001) with CAMHS showing a higher rate (n=14, 51.85%) than paediatrics (n=67, 40.6%) and neurology (n=17, 12.0%).”

We could conclude that the recommendation/golden standard regarding the formation of a multidisciplinary treatment plan, as described in the literature, was not portrayed successfully, when examined the clinical practice in the years 2004-2014 in Denmark.

Reviewer #3 comment #5

What were the reported outcomes of studies that used a multidisciplinary approach, psychoeducation, and psychotherapy in the treatment of FS?

We find that this has been substantially described in discussion section of the study, specifically under 4.1 The multidisciplinary approach and cross-sectoral collaboration, where descriptions of- and outcomes  of studies from Chandra et al, Kozlowska et al., Chudleigh et al., Flewelling et al., and McFarlene et al. have been described.

“Multiple observational studies have shown a decrease in functional seizure frequency, when a multidisciplinary approach is used [20]–[22]. The studies by Chandra et al, Kozlowska et al and Chudleigh et al are based at the same department providing the same 2-week inpatient mind-body program [20], [23], [24]. The creation of teams consisting of somatic healthcare specialists and mental health care specialists were a common denominator in the studies by Kozlowska et al, Chandra et al, Chudleigh et al, Flewelling et al and McFarlene et al [20]–[24]. Similarly, guidance to the patients’ schools was provided and stressors in the patients’ daily lives were identified. However, the treatment in the studies by Flewelling et al and McFarlane et al took place in an outpatient setting [21], [22]. Kozlowska et al found, a greater accept of the diagnosis of FS and improved engagement in treatment at the 12-month follow-up, resulting in 75% of their 60 patients regaining normal function and full-time school attendance [20]. Flewelling et al. also found a decrease in seizure frequency, however, only for 46% of their 19 patients [22]. In total, 58% had improved school attendance and over 50% had a decrease in the usage of emergency departments. Both Flewelling et al. and McFarlane et al. did not find significant improvement regarding anxiety and depressive symptoms [21], [22].”

The following references were used:

[20] Kozlowska, K., Chudleigh, C., Cruz, C., Lim, M., McClure, G. et al. Psychogenic nonepileptic seizures in children and adolescents: Part II – explanations to families, treatment, and group outcomes. Clin. Child Psychol. Psychiatry. 23, 160–176 (2018) doi: 10.1177/1359104517730116. 

[21] McFarlane, F. A., Allcott-Watson, H., Hadji-Michael, M., McAllister, E., Stark, D. et al. Cognitive-behavioural treatment of functional neurological symptoms (conversion disorder) in children and adolescents: A case series. Eur. J. Paediatr. Neurol. 23, 317– 328 (2019) doi: 10.1016/j.ejpn.2018.12.002. 

[22] Flewelling, K. D., Koehler, A., Shaffer, J., Dill, E. J. Medical and psychosocial outcomes of youth with psychogenic nonepileptic seizures: An observational study. Epilepsy Behav. 112 (2020) doi: 10.1016/j.yebeh.2020.107383. 

[23] Chandra, P., Kozlowska, K., Cruz, C., Baslet, G. C., Perez, D. L. et al. Hyperventilation-induced non-epileptic seizures in an adolescent boy with pediatric medical traumatic stress. Harv. Rev. Psychiatry 25, 80–190 (2017) doi: 10.1097/HRP.0000000000000131. 

[24] Chudleigh, C., Kozlowska, K., Kothur, K., Davies, F., Baxter, H.et al Managing nonepileptic seizures and psychogenic dystonia in an adolescent girl with preterm brain injury. Harv. Rev. Psychiatry 21, 163-174 (2016) doi: 10.1177/1359104515621960

Reviewer #3 comment #6

Why were antiseizure medications (ASMs) prescribed to some patients without comorbid epilepsy? What were the potential concerns associated with this practice?

Thank you for question. We hypothesize that patients without epilepsy receive antiseizure medication due to diagnostic uncertainty, however, we do not know the exact reason why some patients without epilepsy received antiseizure medication. Studies have found that antiseizure medication can lead to potential adverse learning and behavioural effects in patients without epilepsy. These points were described in the Discussion section under 4.3 Pharmaceuticals

“Multiple studies have shown that treatment with ASMs is ineffective and can potentially lead to adverse learning- and behavioural effects in patients without epilepsy [4], [28], [29]. The results from the current study could reflect the results found in the study by Plioplys et al [30]. This study suggests that the diagnostic uncertainty by the medical staff contributes to the incorrect treatment with ASMs in patients diagnosed with FS [30].”

The following references were used:

[4] Caplan, R., Doss, J., Plioplys, S., Jones, J. E. Diagnosis of peadiatric PNES in Pediatric Psychogenic Non-Epileptic Seizures (Caplan, R., Doss, J., Plioplys, S., Jones, J. E. eds) 3-14 (Springer International Publishing, Switzerland 2017). 

[28] Buonsenso, D.et al. Paroxysmal non-epileptic events in the pediatric emergency department. Eur. Rev. Med. Pharmacol. Sci. 23, 2188–2193, (2019) doi: 10.26355/eurrev_201903_17265. 

[29] Mandli, A., Desai, N., Badheka, R., Udani, V. Paroxysmal nonepileptic events in a pediatric epilepsy clinic. J. Pediatr. Neurosci. 16, 17-23, (2021) doi: 10.4103/jpn.JPN_33_20. 

[30] Plioplys, S., Siddarth, P., Asato, M. R., Caplan, R. Clinicians’ Views on Antiepileptic Medication Management in Nonepileptic Seizures. 29, 746–750, (2014) doi: 10.1177/0883073813488679

Reviewer #3 comment #7

What were the limitations of the study in terms of data collection and reporting? How might these limitations affect the validity of the findings?

We agree that the limitation section could be further described, hence we used the suggestions provided by reviewer #1, to revise the limitation section. The current section describes how the retrospective design relies on the reporting by health care professionals, which could in turn lead to incomplete data or bias. Furthermore, we acknowledge that human error and subjectivity in data collection is a possibility, since the data extraction was done by a single investigator.

Revision:

“The current study is a retrospective study, and thus relies on the reporting by health care professionals. Insufficient reporting could lead to biases or incomplete data. In addition, the variation in the level of detail in reporting made it difficult to assess the quality of the used modalities. Since the data was extracted by a single investigator, there is a possibility of human error and subjectivity in the data collection and interpretation. The medical records only contained data from the diagnosing department, meaning any service provided at another hospital department e.g., after a referral or at the general practitioners would not be available for review in the current study. Modalities used in the treatment of FS after the first three months post diagnosis of FS was not registered in the current study. The data extracted from the medical records in the current study is from the past decade, why current practice might have changed. Nevertheless, it is important to have a knowledge of previous practice to curate future treatment guidelines”

Reviewer #3 comment 8

What are the implications of the study's findings and recommendations for future treatment guidelines and clinical practice in the field of FS?

This was the first nationwide study to examine the clinical practice regarding the treatment of functional seizures in children and adolescents. We found that a cross-sectoral collaboration and a multidisciplinary approach was somewhat irregularly applied – this is in great contrast to what the literature describes as the golden standard of treatment for functional seizures. Regarding recommendations for future treatment guidelines: This study will aid in the curation of clinical guidelines for the treatment of functional seizures, however the curation of the guidelines is not within the scope of the current study. We find that it is equally important to examine the effect of the current practice before guidelines can be curated.

Reviewer #3 comment 9

Correct the word “behavioural” in the line 304.

We have written the article in British English, hence the word “behavioural” is spelled with an “ou” instead of “o” as in American English.

General revisions:

“Statistical” was changed to “statistic” in two instances in the results section under 3.4 Stratified analysis.

“Assessment” was changed to “assessment in one instance in the results section under 3.4 Stratified analysis.

“Comobidity” was changed to “comorbidity” in one instance in the results section under 3.4 Stratified analysis.

“Speciality” was changed to “specialty” in one instance in the results section under 3.4 Stratified analysis.
